# BUDGET-CONSTRAINED LEARNING TO DEFER FOR AUTOREGRESSIVE MODELS

## ABSTRACT

The learning to defer (L2D) framework gives a model the choice to defer prediction to an expert based on the model's uncertainty. We assume an L2D setting for sequence outputs where a small model can defer *specific outputs* of the whole model prediction to a large model in effort to interweave both models throughout the prediction. We propose a Learn then test approach to tune a token-level confidence-based thresholding rejector for pre-trained predictors with statistical guarantees of being within a user-defined budget and maximizing accuracy. We use Bayesian optimization to efficiently search the space of thresholds. In the experiments, we also empirically demonstrate that this method can achieve budget control while maintaining prediction quality of prediction system in text summarization.

## 1 INTRODUCTION

While large language models have received great praise for their strong predictive capabilities, their inference latency can increase with the size of the model. Recently, Learning to defer (L2D) (Madras et al., 2018) has gained attention for controlling inference costs when there is access to multiple models of various sizes. In L2D, a deferral rule determines whether an input would be routed to a smaller model or a larger model depending on the smaller model's uncertainty on the input.

Existing L2D methods (Mozannar & Sontag, 2020; Verma & Nalisnick, 2022; Mao et al., 2024a;b) are designed to defer the entire prediction to a large model which can quickly become expensive in the case of large sequence outputs. A more cost-efficient approach is to query the larger models only for tokens where the smaller model exhibits high uncertainty. Token-wise deferral can also improve the overall quality of the prediction by interweaving the expert in the small model's prediction, thereby preventing the propagation of model uncertainty. Moreover, current methods do not account for the plausible scenario where the user is operating under a budget, necessitating a strategy that takes into account the limited computational resources and limits the number of queries made to the larger model. If L2D methods are employed in its present form, they will fail to provide guarantees on keeping the rejection costs below a threshold.

To this end, we introduce a deferral rule based on thresholds functions of the token-level uncertainty characterized by the small model. Inspired by Laufer-Goldshtein et al. (2023), we adopt a bayesian optimization (BO) method to efficiently filter the entire search space of thresholds to a subset of promising thresholds that minimize costs while maximizing prediction quality. We then design a *Learn then test* (LTT) framework (Angelopoulos et al., 2021a) to identify the optimal rejector from the subset of thresholds chosen in the previous step, ensuring deferral costs remain within user-specified budget while maximizing prediction quality with high probability. We evaluate the performance of our method with text summarization through which we will assess the quality and the cost of the prediction systems designed by the rejector selected by our method.

## 2 PROBLEM SETTING

Let $\mathcal{V}$ be a vocabulary and $\mathcal{Y} \subseteq \bigcup_{l=1}^{L} \mathcal{V}^l$ be the output space of all sequences with a maximum length of $L$. Let $\mathcal{X}$ represent the feature space. Suppose there are two pre-trained auto-regressive language models $h_1, h_2 : \mathcal{X} \rightarrow \mathcal{Y}$: $h_1$ is a smaller, cost-efficient model, and $h_2$ a larger, more accurate one. Our goal is to use $h_1$ as the baseline and learn when to defer to $h_2$, while ensuring budget adherence.

Influenced by Rayan & Tewari (2025), we adopt a token-level deferral mechanism that leverages the auto-regressive nature of the language models. For each token position $j \in [L]$, we define a quality function $s : \mathcal{X} \times \mathcal{Y}_{\leq j} \to \mathbb{R}$ that evaluates the predicted output. Here, $\mathcal{Y}_{\leq j}$ represents the space of partial sequences up to the $j^{\text{th}}$ token and $s(x, \hat{y}_{1:j})$ could be the softmax output or the entropy of the softmax layer of $h_1$. Additionally, we introduce a threshold $\lambda$ that comes from some domain $\Lambda \subseteq \mathbb{R}$. When generating $j^{\text{th}}$ token, if $s(x, y_{1:j}) > \lambda$, the system defers to the larger model $h_2$ and incurs a cost $c(x, y_{<j})$; otherwise, it maintains the smaller model $h_1$'s prediction. If $\hat{y}_j^{(1)}$ and $\hat{y}_j^{(2)}$ denote the predicted tokens generated by models $h_1$ and $h_2$, respectively, the final output sequence $\hat{y}$ is defined recursively as $\hat{y}_{1:j} = \left( \hat{y}_{<j}, \hat{y}_j^{(1)} \mathbb{1}_{\{s(x, y_{1:j}) \leq \lambda\}} + \hat{y}_j^{(2)} \mathbb{1}_{\{s(x, y_{1:j}) > \lambda\}} \right)$. The total deferral cost incurred by the system for a full prediction would be $\mathcal{L}^{\text{cost}}(\lambda, x, y) = \sum_{j=1}^{L} c\left(x, \hat{y}_{<j}\right) \mathbb{1}_{\{s(x, y_{1:j}) > \lambda\}}$ and the accuracy of the prediction produced by the system is determined by $\mathcal{L}^{\text{acc}}(\lambda, y, \widehat{y})$. Crucially, $\mathcal{L}^{\text{acc}}$ must decrease as the prediction quality improves and it can be selected based on the task.

The risks are denoted as $\mathcal{R}_{\text{cost}}(\lambda) = \mathbb{E}_{(x,y) \sim \mathcal{P}_{X,Y}} \left[ \mathcal{L}^{\text{cost}}(\lambda, x, y) \right]$ and $\mathcal{R}_{\text{acc}}(\lambda) = \mathbb{E}_{(x,y) \sim \mathcal{P}_{X,Y}} \left[ \mathcal{L}^{\text{acc}}(\lambda, y, \widehat{y}) \right]$. Thus, we need to select a threshold $\lambda \in \Lambda$ that minimizes $\mathcal{R}_{\text{acc}}(\lambda)$ while constraining $\mathcal{R}_{\text{cost}}(\lambda)$ to a budget.

Although it would be desirable to directly solve this optimization problem, the lack of oracle access to $\mathcal{P}_{\mathcal{X} \times \mathcal{Y}}$ makes it challenging. Instead, we can use empirical estimates of the risks on a data set to solve a similar optimization problem. If $\hat{\mathcal{R}}_{\text{cost}}^c(\lambda) = \frac{1}{n_c} \sum_{(x^i, y^i) \in \mathcal{D}_c} \mathcal{L}^{\text{cost}}(\lambda, x^i, y^i)$ and $\hat{\mathcal{R}}_{\text{acc}}^c(\lambda) = \frac{1}{n_c} \sum_{(x^i, y^i) \in \mathcal{D}_c} \mathcal{L}^{\text{acc}}(\lambda, x^i, y^i)$ are the empirical estimate of $\mathcal{L}_{\text{cost}}$-risk and $\mathcal{L}_{\text{acc}}$-risk respectively using the calibration data set $\mathcal{D}_c = \{(x^i, y^i)\}_{i=1}^{n_c}$, we aim to identify $\widehat{\lambda}$ that satisfies the following $(\alpha, \delta)$ risk-controlling property such that

$$\mathbb{P} \left( \mathcal{R}_{\text{cost}} \left( \widehat{\lambda} \right) \leq \alpha \right) \geq 1 - \delta \tag{1}$$

where the probability is with respect to $\mathcal{D}_c$ and $\alpha$ is the user-defined deferral budget. If a set of configurations $\widehat{\Lambda}$ satisfy this property, $\widehat{\lambda}$ will be the maximizer of $\hat{\mathcal{R}}_{\text{acc}}^c(\lambda)$ among $\widehat{\Lambda}$.

## 3 METHODOLOGY

To achieve the risk control described in Equation 1, a grid search over $\Lambda$ is typically employed. However, evaluating $\hat{\mathcal{R}}_{\text{acc}}^c(\lambda)$ and $\hat{\mathcal{R}}_{\text{cost}}^c(\lambda)$ on every grid point can be computationally expensive and the quality of the search would rely on the precision of the discretization of the domain. Additionally, the costs of exploring the grid are further exacerbated by the dimensionality of the sequence length. Inspired by Laufer-Goldshtein et al. (2023), we deploy hyperparameter optimization (HPO) techniques based on Bayesian Optimization (BO) (Wang et al., 2022) to identify thresholds that minimize cost and maximize accuracy while efficiently exploring the search space. These thresholds will then be inputted into the Learn Then Test (LTT) framework (Angelopoulos et al., 2021a) to identify $\widehat{\lambda}$ with high-probability guarantees of budget adherence while maintaining high accuracy. This approach enables us to direct the LTT method towards regions of the search space where a risk-controlling $\lambda$ is more likely to be identified instead of uniformly searching the entire domain.

### 3.1 HYPERPARAMETER OPTIMIZATION (HPO)

Uncovering relevant parts of the search space requires paying heed to the cost-accuracy trade off: improving accuracy typically incurs higher computational costs as a result of increased large model queries. Therefore, no single solution optimizes both risks simultaneously. Instead, we aim to discover a Pareto-optimal set which is defined as:

$$\Lambda_{\mathcal{P}} = \left\{ \lambda \in \Lambda \mid \nexists \lambda' \in \Lambda \text{ with } \hat{\mathcal{R}}_{\text{acc}}^c(\lambda') \leq \hat{\mathcal{R}}_{\text{acc}}^c(\lambda), \ \hat{\mathcal{R}}_{\text{cost}}^c(\lambda') \leq \hat{\mathcal{R}}_{\text{cost}}^c(\lambda), \lambda' \neq \lambda \right\}$$

Intuitively, the Pareto-optimal set contains configurations that cannot be outperformed by any other in terms of both accuracy and cost. Thus, the Pareto front can be constructed as follows - $\mathcal{P} := \{(\hat{\mathcal{R}}_{\text{acc}}(\lambda), \hat{\mathcal{R}}_{\text{cost}}(\lambda)) \mid \lambda \in \Lambda_{\mathcal{P}}\}$. To evaluate the quality of a Pareto front, $\mathcal{P}$, we use the Hypervolume

Indicator (HV) (Deb, 2001), defined with respect to a reference point $r \in \mathbb{R}^2$

$$HV\left(\mathcal{P}, r\right) = \int_{\mathbb{R}^2} \mathbb{1}\left\{z \in H\left(\mathcal{P}, r\right)\right\} dz$$

where $H\left(\mathcal{P}, r\right) = \left\{z \in \mathbb{R}^2 \mid \exists p \in \mathcal{P} : p_1 \leq z_1 \leq r_1, p_2 \leq z_2 \leq r_2\right\}$. The Hypervolume indicator measures the volume in the objective space dominated by a set of Pareto-optimal solutions with respect to a reference point. Typically, the reference point is set to the nadir of the Pareto front, representing the worst objective values in the approximation to ensure that all Pareto optimal solutions have positive hyper-volume contributions. To identify efficient thresholds that balance cost and accuracy, we pursue the maximization of Hypervolume Improvement (HVI):

$$HVI\left(\lambda, \mathcal{P}; r\right) = HV\left(\mathcal{P} \cup (\hat{\mathcal{R}}_{\text{acc}}^c(\lambda), \hat{\mathcal{R}}_{\text{cost}}^c(\lambda)), r\right) - HV\left(\mathcal{P}, r\right) \tag{2}$$

Maximizing the HVI helps us draw nearer to finding the optimal balance between cost and accuracy. Given the computational expense of direct optimization of Equation (2), we approximate each risk using Gaussian Process (GP) surrogates. At each iteration, the posterior mean of the GP is used to compute Equation (2), guiding the selection of the next candidate hyperparameter $\lambda$. The true empirical risks $\hat{\mathcal{R}}_{\text{acc}}^c(\lambda), \hat{\mathcal{R}}_{\text{cost}}^c(\lambda)$ of the candidate are then used to refit the GPs. This procedure is iterated until the maximum number of iterations is reached, resulting in an approximate Pareto front. The complete pseudo-code of this procedure and a visual representation of HVI can be found in Appendix B.

## 3.2 LEARN THEN TEST (LTT)

Having obtained an optimal search space $\Lambda_{\mathcal{P}}$ from the HPO step, we can now use the "Learn then test" approach to select a threshold with statistical guarantees. LTT (Angelopoulos et al., 2021a) aims to calibrate learned models to provide finite-sample guarantees without assuming the underlying data distribution. LTT seeks to upper bound the expectation of general losses conditioned on $\mathcal{D}_c$ with high probability. It starts with a predictor $f$ which is trained on some data set $\mathcal{D}_t$ and it uses $\mathcal{D}_c$ data to learn a $\hat{\lambda}$ using a multiple testing procedure such that the interleaved prediction $\hat{y}$ which controls the risk $\mathcal{R}_{\text{cost}}(\lambda)$ at the level $\alpha$ with probability $1 - \delta$.

To apply LTT for token-level deferrals, for a given budget $\alpha$ and error level $\delta$, we must: 1) find finite sample and super-uniform p-values $p_\lambda$ for the null hypothesis $H_\lambda : \mathcal{R}_{\text{cost}}\left(\lambda\right) > \alpha$ for each $\lambda \in \Lambda$ using $\widehat{\mathcal{R}}_{\text{cost}}\left(\lambda\right)$, 2) apply a family-wise error rate (FWER) controlling multiple hypothesis strategy to identify a subset of non-rejected values of lambda or $\widehat{\Lambda}$, 3) select $\hat{\lambda}$ with the smallest $\widehat{\mathcal{R}}_{\text{acc}}\left(\lambda\right)$ from $\widehat{\Lambda}$. For the first step, we use Lemma 3.1 to calculate p-values for each $\lambda \in \Lambda_{\mathcal{P}}$. The proof of Lemma 3.1 is adapted from a similar proof in (Quach et al., 2023), and we defer the proof to Appendix C.

**Lemma 3.1** *If $Binom(n, p)$ is a binomial random variable with $n$ trials and success probability $p$, then $p_\lambda^c = \mathbb{P}\left(Binom(n_c L, \alpha) \leq n_c L \widehat{R}_{cost}^c(\lambda)\right)$ is a valid p-value, where $\widehat{R}_{cost}^c(\lambda)$ is the empirical $\mathcal{L}_{cost}$-risk on $\mathcal{D}_c$*

With $\{p_\lambda^c\}_{\lambda \in \Lambda_{\mathcal{P}}}$, we can pass the p-values into a multiple hypothesis testing algorithm. While the Bonferroni procedure is a valid FWER controlling algorithm, the multiplicity correction tends to degrade the power of the test as $|\Lambda_{\mathcal{P}}|$ increases, resulting in $\widehat{\Lambda} = \emptyset$. Fixed sequence testing is optimal when we have apriori information about the likelihood of rejecting $H_\lambda$. To collect this information, we employ Pareto testing (Laufer-Goldshtein et al., 2022).

In Pareto testing, the Pareto front described in Section 3.1 is filtered out of $\Lambda_{\mathcal{P}}$ and the Pareto front is ordered by increasing p-values from Theorem 3.1 using $\mathcal{D}_c$. P-values $p_\lambda^d$ are then computed using the empirical $\mathcal{L}_{\text{cost}}$-risk on another dataset $\mathcal{D}_d$ or $\widehat{R}_{\text{cost}}^d(\lambda)$. Finally, fixed sequence testing is applied in the order specified in the previous step to return a set of acceptable configurations or $\widehat{\Lambda}$. If $|\widehat{\Lambda}| > 1$, we select the configuration with the lowest empirical $\mathcal{L}_{\text{acc}}$-risk on $\mathcal{D}_d$ or $\widehat{R}_{\text{acc}}^d(\lambda)$.

**Theorem 3.2** *If $\hat{\lambda} = \underset{\lambda \in \Lambda_{accept}}{\operatorname{argmin}} \widehat{R}_{acc}^d(\lambda)$, $(\alpha, \delta)$ risk control or Equation (1) has been satisfied.*

The proof of Theorem 3.2 follows from Theorem 1 in (Angelopoulos et al., 2021b).

## 4 EXPERIMENTS

We perform our experiments on the *Extreme Summarization* task to demonstrate the performance of a confidence thresholding rejector. $h_1$ or the small model is a *t5-small* model (Niemiec) finetuned on the XSUM dataset (Narayan et al., 2018) while $h_2$ is a *t5-large* model (Stept) finetuned on both XSUM and CNN Daily Mail (Nallapati et al., 2016) data sets. For the calibration procedure, 400 article-summary pairs from the XSUM dataset whose summary length doesn't exceed 20 tokens are selected. 200 of those samples form $\mathcal{D}_c$, which is used for BO and the first stage of Pareto testing step while the rest or $\mathcal{D}_d$ is used for the second stage of Pareto testing. The models were set to generate sequences with maximum length of 20 tokens. $\widehat{\lambda}$ is evaluated against 100 test points with summary lengths less than 20 tokens with the results averaged over 5 runs.

Here, $q(x, \widehat{y}_{<j})$ is the negative logarithmic softmax output of the greedy token prediction from $h_1$. The deferral cost $c(x, \widehat{y}_{<j})$ is set to 0.05, making $\mathcal{L}_{\text{cost}}$ the fraction of expert calls per prediction. We used $1 - \text{ROUGE}$ score as $\mathcal{L}_{\text{acc}}$ to evaluate prediction quality. A Matern kernel with $\nu = 2.5$ is used to fit the GPs. LTT is implemented with various $\alpha$ levels upto 0.2 and $\delta = 0.1$. All inference passes are done on an Nvidia A40 GPU. We evaluate $\widehat{\lambda}$ on total deferral cost $\mathcal{L}_{\text{cost}}$ and prediction quality $\mathcal{L}_{\text{acc}}$. We test how often the L2D system stays within budget and achieves the desired prediction quality for various $\alpha$ levels on a test set.

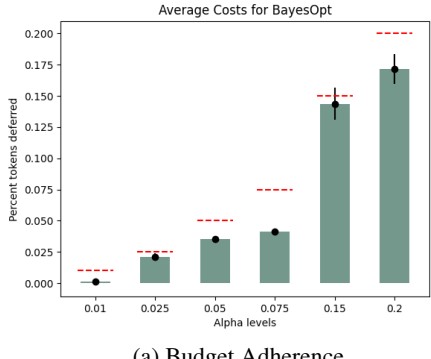
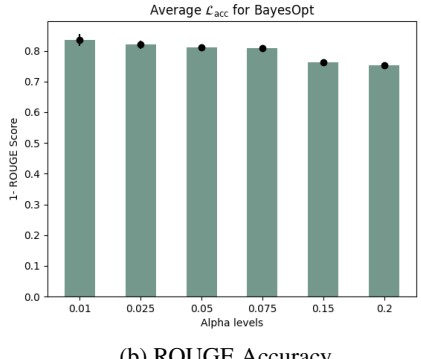

(a) Budget Adherence          (b) ROUGE Accuracy

Figure 1: Figure (a) plots the average cost against various alpha levels with the red dotted line denoting the budget or $\alpha$ level. Figure (b) is a plot of the ROUGE accuracy.

We demonstrate how our method performs in terms of accuracy and budget adherence. Figure 1a demonstrates how our method complies with the budget across all alpha levels. For certain budget settings, our method tends to be too conservative even when exhausting the budget would better maximize accuracy and balance the cost-accuracy trade off. However, in Figure 1b, we see that using BO gives increasing accuracy as the budget increase.

## 5 DISCUSSION

This work raises many interesting questions. The choice of having a uniform threshold for all token positions was made to provide ease in implementing the BO step. To make the thresholds multidimensional without BO suffering from curse of dimensionality, we would like to implement *group thresholds*. For group thresholds, a single threshold will be used for contiguous portions of the sequence generation as opposed to the whole sequence. Furthermore, we are also interested in using other state-of-the-art HPO methods like HyperBand (Li et al., 2018) and Bayesian optimization hyperband (Wang et al., 2018). We expect using methods that perform better than BO in HPO can select better threshold configurations for hypothesis testing procedure.

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

# A  RELATED WORK

**Cascades and Learning to Defer.** Cascading is a line of work that involves routing input to the right models. This is often done by thresholding the predicted probabilities (Wang et al., 2017; Narasimhan et al., 2024; Jitkrittum et al., 2024). Our method most closely aligns with these methods. However, they lack guarantees on budget adherence and simply maximize prediction quality. Specifically, Narasimhan et al. (2024) queries both models to determine deferral decisions, defeating the purpose of this framework. Inspired by learning to defer methods (Mozannar & Sontag, 2020; Verma & Nalisnick, 2022), some cascade approaches use deep learning architectures to model a rejector (Wang et al., 2023; Ding et al., 2024; Wang et al., 2024; Gupta et al., 2024), however their training procedures for whole deferrals don't feasibly extend to token-wise deferrals and lack nice statistical properties that typical learning to defer loss functions possess.

**Calibration Methods.** This deferral problem aims to characterize token-wise model uncertainty with guarantees on its accuracy. This closely relates problems tackled by calibration methods. Calibration concerns processing predictions to account for model uncertainty providing some statistical guarantees. There is a plethora of work (Ren et al., 2023; Deutschmann et al., 2024; Ravfogel et al., 2023) that uses conformal prediction in language models to provide coverage guarantees. However, these methods focus on capturing overall sequence-level uncertainty, making it difficult to glean granular insights. Quach et al. (2023) uses the "Learn then test" framework (Angelopoulos et al., 2021a) to achieve risk control. However, they perform a grid search over a high-dimensional space which can be computationally intractable. Our method is inspired by Laufer-Goldshtein et al. (2023) which replaces the grid search with a bayesian optimization to efficiently control multiple risks.

# B  ALGORITHM

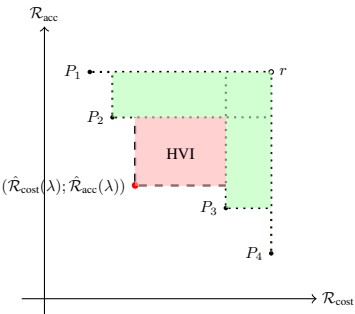

Figure 2: Illustration of the HVI with the reference point $r$ set to the nadir. The green-shaded region represents the HV of the Pareto front $\{P_1, P_2, P_3, P_4\}$. The red-shaded rectangle highlights the additional hypervolume contributed by the new point $(\hat{\mathcal{R}}_{\text{cost}}(\lambda), \hat{\mathcal{R}}_{\text{acc}}(\lambda))$, denoting the HVI.

---

**Algorithm 1:** Approximate Pareto Front with BO

**Input:** Initial set of configurations $\Lambda_0$, Calibration set $\mathcal{D}_c$
**Output:** Optimized set of configurations $\Lambda_{\mathcal{P}}$

Initialize $\Lambda_{\mathcal{P}} \leftarrow \Lambda_0$;
**repeat**

 Fit Gaussian Processes for $\hat{\mathcal{R}}_{\text{acc}}^c$ and $\hat{\mathcal{R}}_{\text{cost}}^c$ using $\Lambda_{\mathcal{P}}$;
 Compute the Pareto front $\mathcal{P}$ of $\Lambda_{\mathcal{P}}$;
 Compute the reference point $r$;
 Maximize Hypervolume Improvement (HVI) acquisition function to propose new $\lambda$;
 Calculate accuracy risk $\hat{\mathcal{R}}_{\text{acc}}^c(\lambda)$ and cost risk $\hat{\mathcal{R}}_{\text{cost}}^c(\lambda)$ on $\mathcal{D}_c$;
 Update $\Lambda_{\mathcal{P}} \leftarrow \Lambda_{\mathcal{P}} \cup \{\lambda\}$;

**until** *maximum iterations*;
**return** $\Lambda_{\mathcal{P}}$;

---

## C  PROOF OF THEOREM 3.1

We will show that $p_\lambda^c$ is super uniform or $\forall u \in [0, 1]$ $\mathbb{P}(p_\lambda^c \leq u) \leq u$ conditioned on the null hypothesis being true. Let $A = \text{Binom}(n_cL, \alpha)$ and let $Y = n_cL\widehat{\mathcal{R}}_{\text{cost}}(\lambda)$. Let's say $Y \sim B = \text{Binom}(n_cL, \alpha')$. Under the null hypothesis, $R(T_\lambda) > \alpha$. We know that $\widehat{\mathcal{R}}_{\text{cost}}(\lambda)$ is an unbiased estimate of $\mathcal{R}_{\text{cost}}(\lambda)$ or $\mathbb{E}\left[\widehat{\mathcal{R}}_{\text{cost}}(\lambda)\right]$ because we are just taking an empirical estimate. The null hypothesis can be restated as follows:

$$\alpha < \mathcal{R}_{\text{cost}}(\lambda)$$
$$= \mathbb{E}\left[\widehat{\mathcal{R}}_{\text{cost}}(\lambda)\right]$$
$$= \alpha'$$

So under the null hypothesis, I can say that

$$F_B(z) \leq F_A(z)$$

Now the p-value can be rewritten as:

$$\mathbb{P}\left(\text{Binom}(n_cL, \alpha) \leq n_cL\widehat{\mathcal{R}}_{\text{cost}}(\lambda)\right) = P(A \leq B) = F_A(B)$$

So, for $u \in [0, 1]$

$$\mathbb{P}(p_\lambda < u) = \mathbb{P}(F_A(B) < u)$$
$$\leq \mathbb{P}(F_B(B) < u) \text{ under the null, } \alpha' > \alpha$$
$$= \mathbb{P}\left(B < F_B^{-1}(u)\right)$$
$$= F_B(F_B^{-1}(u))$$
$$= u$$

