# OpenReview forum: "Budget-Constrained Learning to Defer for Autoregressive Models"
_ICLR.cc/2025/Workshop/BuildingTrust — Submitted to BuildingTrust_

### Official Review · Reviewer_GGrW · 2025-02-27
**Budget-Constrained Learning to Defer for Autoregressive Models**

**Rating:** 6
**Confidence:** 2

**Review:**

The paper presents a well-structured methodology, systematically dividing the approach into Hyperparameter Optimization (HPO) and the Learn Then Test (LTT) framework, making it easy to follow. The use of Gaussian Processes and Hypervolume Indicators (HVI) for threshold selection is an effective strategy for efficiently navigating the optimization space. However, the results lack sufficient explanation, which diminishes the novelty and impact of the findings. While the study emphasizes cost efficiency, there is little discussion on the actual computational cost compared to grid search or simpler heuristic approaches, making it unclear how to benchmark the efficiency of this method. Additionally, deferral decisions are based on softmax entropy, yet softmax probabilities are known to be overconfident, and the paper does not discuss how bias in the smaller model might affect deferral performance. Another critical consideration is the curse of dimensionality in Bayesian Optimization, which is briefly mentioned in the discussion. The study sets a maximum sequence length of 20 tokens, but real-world applications often require handling significantly longer sequences. As sequence length increases, the dimensionality of the BO problem grows, making it more difficult for Gaussian Processes (GPs) to generalize effectively. A Survey on High-Dimensional Gaussian Process Modeling with Application to Bayesian Optimization (Binois & Wycoff) might help, this paper highlights that GPs struggle in high-dimensional spaces due to distance concentration effects and the exponential increase in complexity. Addressing scalability concerns and potential mitigation strategies, such as dimensionality reduction techniques or structured optimization, would significantly enhance the study’s practical applicability.

---

### Official Review · Reviewer_mDP2 · 2025-03-02
**Mathematically sound and Interesting Methodology, Lacking Relevance to Trust**

**Rating:** 6
**Confidence:** 2

**Review:**

# Evaluation of the Work

## Summary of my review
Overall, the paper is interesting and mathematically well thought out. It represents an interesting approach. However, I do not see a link between this kind of learning method and trust in LLMs. I question the relevance of this work to the workshop while appreciating its general validity. I finally settled on a weak accept, as I do not believe it is a fit in the workshop topics presented (or anything related to trust), but I do not know what the workshop curators define as other relevant categories.

## Quality
**Pros:**
- Mathematically justified and sound method
- Tested and validated methodology

**Cons:**
-  Limited testing. It feels as though the extreme summarization task is far too limited for the proposal. However, being a workshop paper, I understand that this methodology is likely still under test.
- No justification as to why extreme summarization is an acceptable/appropriate task. I am left with questions as to why you chose this task and why it is representative of the field.

## Clarity
**Pros:**
-  As an applied ML researcher, I was able to understand the difficult mathematical equations and the explanation of the methodology
- The introduction lays out the exact contribution of the paper, as well as the layout of the paper

**Cons:**
-  Perhaps for a Trust workshop, more time should be spent on emphasizing how this model conforms to the trust criteria laid out in the description
- I did not understand Figure 1b. To me, it looks a though accuracy decreases as the budget increases. This directly contradicts lines 200-204, explaining the figure.
- Occasionally, I felt that the explanations of the methodology, particularly those around hyper-parameter tuning, were too in-depth, while ignoring the aspects of the paper (Discussion and Experiment being 1 page with a figure, compared to the 4 pages spent explaining the methodology)

## Originality
As much as I would like to comment on the originality, I have never worked with BO. I do not have a substantial enough grasp of the literature. From my limited understanding, it seems relatively novel but mostly builds upon the LTT framework. Essentially, the point of the work is a justification for using the LTT framework more effectively.

## Significance & Relevance
In a broad sense, it falls under "Error detection and correction," perhaps? However, I can not find any workshop topic that directly contributes to it. The paper, while significant in its own way, does not mention trust or LLM robustness. It overlooks the purpose of the workshop. I was hopeful that the discussion would explain the relationship of this paper to trust, but instead, it focused on future research directions.

---

### Decision · Program_Chairs · 2025-03-04

**Decision:**

Reject

**Comment:**

There is no comparison with Speculative Decoding where the large model can simply abstain based on the smaller model's uncertainty. The novelty of this work is limited to the best of my understanding given the well-known SD framework